# Approaches used to prevent and reduce the use of restrictive practices on adults with learning disabilities: Protocol for a realist review

Alina Haines-Delmont[1]*, Anthony Tsang[1], Kirstine Szifris[2], Elaine Craig[1], Melanie Chapman[3], John Baker[4], Peter Baker[5], James Ridley[6], Michaela Thomson[7], Gary Bourlet[8], Beth Morrison[9], Joy Duxbury[1]

1 Department of Nursing, Manchester Metropolitan University, Manchester, United Kingdom, 2 Policy Evaluation and Research Unit, Manchester Metropolitan University, Manchester, United Kingdom, 3 Department of Social Care and Social Work, Manchester Metropolitan University, Manchester, United Kingdom, 4 School of Healthcare, University of Leeds, Leeds, United Kingdom, 5 Social Policy, Sociology and Social Research, University of Kent, Kent, United Kingdom, 6 Medical School, Edge Hill University, Liverpool, United Kingdom, 7 Mersey Care NHS Foundation Trust, Prescot, United Kingdom, 8 Learning Disability England, Birmingham, United Kingdom, 9 Positive & Active Behaviour Support Scotland, Monifieth, United Kingdom

ʘ These authors contributed equally to this work.
* a.haines@mmu.ac.uk

## Abstract

### Introduction

The use of restrictive practices has significant adverse effects on the individual, care providers and organisations. This review will describe how, why, for whom, and in what circumstances approaches used by healthcare organisations work to prevent and reduce the use of restrictive practices on adults with learning disabilities.

### Methods and analysis

Evidence from the literature will be synthesised using a realist review approach - an interpretative, theory-driven approach to understand how complex healthcare approaches work in reducing the use of restrictive practices in these settings. In step 1, existing theories will be located to explore what approaches work by consulting with key topic experts, holding consultation workshops with healthcare professionals, academics, and experts by experience, and performing an informal search to help develop an initial programme theory. A systematic search will be performed in the second step in electronic databases. Further searches will be performed iteratively to test particular subcomponents of the initial programme theory, which will also include the use of the CLUSTER approach. Evidence judged as relevant and rigorous will be used to test the initial programme theory. In step three, data will be extracted and coded inductively and deductively. The final step will involve using a realist logic of analysis to refine the initial programme theory in light of evidence. This will then provide a basis to describe and explain what key approaches work, why, how and in what

relevant data from this study will be made available upon study completion.

**Funding:** Study funded by the National Institute for Health Research Health Services and Delivery Research (NIHR HS&DR). Ref: NIHR129524. URL: https://fundingawards.nihr.ac.uk/award/NIHR129524 Authors who have received the award: AH-D, JD, EC, PB, JB, MC, BM, JR, KS, MT, GB. The views and opinions expressed in the paper are those of the author(s) and not necessarily reflect those of the HS&DR Programme, NIHR, NHS or the Department of Health. YES - the study was reviewed by the NIHR HS&DR part of competitive funding and reviewers' comments were taken into account in the re-drafting of the protocol. the funders did not have a role in preparation of this manuscript. Any publication emerging from the research needs to be notified on the NIHR award system.

**Competing interests:** The authors have declared that no competing interests exist.

circumstances in preventing and reducing the use of restrictive practices in adults with learning disabilities in healthcare settings.

## Results

Findings will be used to provide recommendations for practice and policymaking.

## Registration

In accordance with the guidelines, this realist review protocol was registered with the International Prospective Register of Systematic Reviews (PROSPERO) on 4[th] December 2019 (CRD42019158432).

## Introduction

### Rationale

There are approximately 1.5 million individuals with a learning disability in the UK [1] and up to 60-70% of this population are autistic [2]. Individuals with a primary diagnosis of a learning disability are more likely to have a wide-ranging number of physical and mental comorbidities that include schizophrenia, epilepsy, depressive disorders, hearing loss, and visual impairment [3]. Adults with increased severity of a learning disability and the presence of communication difficulties have been found to be consistently associated with a higher risk of displaying behaviour that challenges [4–6]. Empirical evidence indicates that the presence of behaviour that challenges is the most prominent characteristic that is linked with incidents of restrictive practices such as restraint, rapid tranquilisation, and seclusion in these healthcare settings [7–9].

Despite global consensus to prevent and reduce the use of these controversial practices, these are still commonly used in inpatient and community settings for people with learning disabilities [10–12]. Recent evidence has demonstrated more than a 50% increase in the use of such practices on adults with a learning disability in hospitals in England from 2016 to 2017 [13]. The consequences of the use of restrictive practices can result in significant trauma for patients, physical injuries and burnout for staff, frustration and reduced quality of life for carers [11, 14–16].

The most common approaches used to prevent or reduce the use of restrictive practices in learning disability settings are centered around Positive Behaviour Support (PBS) [17–21] underpinned by a person-centered, trauma informed approach. These may also cover the implementation of behaviour support plans (BPSs) [22, 23]; staff training in mindfulness/Mindfulness-Based Positive Behavior Support (MBPBS) [24–26]; programmes [27] including elements of PBS, Safewards [28] and the Six Core Strategies [29] or organisational behavior management (OBM) approaches to reducing the use of restrictive practices in these settings [30].

Gaskin et al.'s systematic review [10] of 14 single-subject design studies evaluating interventions targeting the reduction of use of restrictive practices such as physical and mechanical restraint on people with developmental disabilities identified a mean reduction in frequency of restraint of over 70% between the baseline and intervention phases. Three types of restraint reduction approaches were reported: (1) those targeting the reduction of restraint with people displaying agitation or aggressive behaviour (e.g. medication to enable night-time sleeping or

other medication changes; antecedent assessment and modifying antecedent conditions and behaviour-specific criteria for restraint; involving patients in behavioural support plans); (2) those targeting the reduction of restraint with people who self-harmed (e.g. offering choice to patients regarding staff to work with; fixed time release from restraint; behavioural assessment and treatment; training involving relaxation, increasing time out of restraint, using hands for other activities); (3) those taking an organisation-wide restraint reduction stance (e.g. training on reducing aggression; behavioural training for staff; mindfulness training; organisational behaviour management including the use of behavioural plans, data informed practice and contingencies for mechanical restraint). The results were promising for both instances where restraint was used to manage aggression and self-harm, suggesting that it is achievable to reduce the use of restrictive practices, even if it is not always clear which intervention influences which outcome and why (given the design limitations and the complexity of these settings). The most successful approaches were the organisation-wide initiatives. Gaskin argues that a key limitation is the lack of evidence with regards to large scale, multi-component organisation-wide approaches to reduce restrictive practices in these settings, which is more common in the mental health literature [31–33].

The positive results from Gaskin's review are in line with those reported in Luiselli's earlier review [34] of single-case and small group studies evaluating the implementation of antecedent intervention procedures and fixed time release contingencies to reduce the use of physical restraint for people with intellectual disabilities in community settings. The first approach implies the assessment and change of circumstances surrounding/associated with restraint, the second limits the duration of restraint by using a fixed-time release (FTR) approach [34]. More recently, Sturmey et al.'s systematic review [35] concludes that the most effective approach to date in group restraint reduction is mindfulness, although more research is needed to strengthen the evidence, as well as to identify the mechanisms of change (p.387).

The disparity between existing guidelines and policies to reduce the use of restrictive practices on people with learning disabilities and actual points to the need to develop effective approaches to minimise the use of these practices as well as gather and disseminate the evidence in such a way to enable change in practice. Although the evidence above supports the use of various approaches to reduce restrictive practices in settings providing care for people with learning disabilities and autism, there is a knowledge gap of how and why such approaches work and in what contexts. Using a realist review methodology will help us unpick some of the underlying processes/mechanisms that generate the desired outcomes. Additionally, integrating the views of people with lived experience (patients and carers) will help us identify new mechanisms and enrich and improve our understanding of existing evidence. Involving people with lived experience, especially carers, is something that is lacking both in primary and secondary research in this area.

## Methods

### Aim

The aim of this realist review is to understand how, why, for whom, and in what circumstances approaches used by healthcare organisations work to prevent and reduce the use of restrictive practices on adults with learning disabilities. This will help improve policy and practice in this area.

### Design: A realist approach

Realist review is an interpretative, theory-driven approach that permits the synthesis of an array of evidence types including qualitative, quantitative and mixed-methods research [34].

Realist methodology recognises how and why context influence outcomes. It is understood that particular contexts trigger mechanisms that generate certain outcomes; by providing a narrative based on the evidence of what is most likely to work, how and when [35].

A realist approach was chosen as one of the main strengths is its capacity to recognise and manage the complexity and heterogeneity of approaches used to prevent and minimise the use of restrictive practices. Instead of focusing on what approaches are used or their effectiveness, a realist review interrogates how these approaches, or their components, produce intended outcomes. The refined programme theory will be supported by substantive theory and expressed at the middle range level. This means the theory will be sufficiently broad to allow for transferability of findings to inform the design and implementation of approaches used across different settings [36, 37].

The process of generative causation is iterative and starts with the development and refinement of a realist programme theory of multi-faceted approaches or interventions to prevent or minimise the use of restrictive practices in adults with learning disabilities. To achieve this, an informal search of the literature and consultation with stakeholder groups will help identify the key approaches that are used. The scope will be purposively broad to permit exploration of key approaches. Overlapping components will be homogenised and grouped into conceptual labels that will facilitate data coding. For each conceptual label, a realist logic of analysis will be applied to provide an explanatory account of how the interaction between contexts and mechanisms lead to outcomes. For each conceptual label, mechanism(s) generating certain outcome (s) will be identified and in what contexts these mechanisms may be triggered [38, 39].

In this review, contexts are defined as pre-existing structures that modify and/or trigger the behaviour of mechanisms [40]. Mechanisms are underlying processes or structures that are sensitive to the variation in context, they generate outcomes, and are usually hidden [41].

The realist review protocol has been prospectively registered with PROSPERO [42]. The review will adhere to current RAMESES quality and publication standards [43] and is expected to run for a 22-month period from September 2020. The following steps are informed by Pawson's iterative approach [44, 45].

## Step 1: Locating existing theories

The purpose of this initial step is to develop an initial programme theory that will be used as the basis to conduct a systematic search of literature. This will involve exploring what healthcare approaches are currently in use and are deemed to work in preventing and reducing restrictive practices within learning disabilities settings, how different components are thought to have caused this, and the pre-existing structures in place for this to occur. This will include attempting to identify theories that underpin why certain components are required within existing interventions to achieve desired outcomes. Within such theories, there may be explanations and reasonings with which how an intervention was developed (e.g. who designed it and how?) as these may affect outcomes.

To identify key approaches and theories, the project team will first: i) consult with key topic experts part of the project research management and advisory groups; ii) hold a number of consultation workshops with academic experts, experts by experience, and healthcare professionals that work with people with learning disabilities; and iii) informally search the literature. This scoping search differs from the comprehensive, formal searching process that follows later (Step 2). It is designed to be exploratory, with the view to identify the range of possible approaches and explanatory theories that may be considered relevant. The initial programme theory will be developed from these sources to be tested in the review. Iterative discussions within the project team will be required to build and make sense of approaches used into an

initial plausible and coherent programme theory. Content experts from the wider team will be consulted for programme theory refinement.

## Step 2: Searching for evidence

The aim of this step is to identify a body of literature that contains relevant data to further develop and test the initial programme theory developed from Step 1. The search strategy will be structured and guided by the initial programme theory, previous relevant reviews [10, 33, 46, 47] and by consultation with the project stakeholder groups (i.e. research team, advisory panel, and experts by experience groups). The initial comprehensive search will focus on evidence published since 2001 up to July 2021, to align with the publication of a key policy document - "Valuing People A New Strategy for learning Disability for the 21st Century" – a White Paper setting out the UK Government's commitment to change practice with the view to improve the life chances of people with learning disabilities [48].

Searches will be reported in line with PRISMA-S 2021 guidelines [49] and the following electronic databases will be used seek for relevant evidence: ASSIA (ProQuest), CINAHL (EBSCOhost), Embase (Ovid), Medline (Ovid), PsycINFO (Ovid) and Web of Science Core Collection using one citation index (Emerging Sources Citation Index [2015-present]). Search strategies will be adapted for different databases as required. Where applicable, the CLUSTER searching approach will be employed throughout each iteration of searches. The CLUSTER approach provides a systematic framework for supplementary searching that draws on well-established retrieval practices [50]. CLUSTER complements the iterative and non-linear searches in realist reviews that strongly rely upon the identification of theory [51]. A free hand search on ProQuest and OpenGrey will be conducted if the CLUSTER technique yields insufficient grey literature. Additional sources will be identified via topic experts, healthcare professionals, and experts by experience for any useful websites or organisations to contact, if necessary.

For the initial comprehensive search, the eligibility criteria will be deliberately broad as quantitative, qualitative, mixed-methods, and unpublished evidence will be considered. For the purposes of this review, restrictive practices will be defined as "deliberate acts on the part of other person(s) that restrict an individual's movement, liberty and/or freedom to act independently" [52]. This will include practices such as observation, seclusion and long-term segregation, and all forms of restraint (e.g. physical, mechanical and chemical) [53]. The full eligibility criteria will be fully defined following the completion of Step 1, including consultation with stakeholders.

The following indicative inclusion criteria will be applied: i) all study designs; ii) adults (≥18 years old) with a diagnosis of learning disabilities (i.e. impaired intellectual and social functioning abilities) who may also have a diagnosis of autism or mental health problems (e.g. schizophrenia, anxiety disorders and depression); iii) all healthcare settings; iv) all approaches or interventions that focus on preventing or reducing the use of restrictive practices; and v) all restrictive practice related outcome measures (e.g. reduction in rate of restraint or seclusion). Studies will be excluded based on the indicative subsequent criteria: i) pharmacological (i.e. non-behavioural) interventions and ii) when outcome data of interest for adults cannot be disaggregated from non-adults (i.e. <18 years old).

Studies will be selected for analysis and synthesis based on relevance and rigour [34]. Relevance pertains to whether a study can contribute to programme theory building and/or testing, and rigour is whether the methods used to generate the relevant data are considered credible and trustworthy. Relevance of articles will be categorised into low and high relevance. Articles from the main search will be considered as lower relevance when their findings were not

specific for the target group of this review (i.e., adults with a diagnosis of a learning disability). For instance, articles from the main search will be categorised as being of lower relevance when: i) learning disability was not the primary population of study or less than 50% of the population within the study had a learning disability diagnosis and ii) approach used in study to target the reduction of restrictive practices lacked transparency to allow for replication. At the point of categorising relevance, the rigour of each article will also be examined. For example, if data had been generated by methods that had been clearly explained and justified, then the rigour of data will be considered to be greater if methodology used had not been explained or justified. This approach will be adopted for two reasons. It is anticipated that the searches will yield opinion pieces, editorials and other forms of evidence that cannot be appraised using traditional quality assessment tools. Also, evidence that may meet the full eligibility criteria, but still may not contain any relevant data for the purposes of developing and refining the initial programme theory.

Search results will be imported into the online systematic review management software Covidence. Eligibility of evidence will be undertaken independently by two reviewers at title/abstract and at full-text stage. Any disagreements will be resolved by discussion. If any ambiguities still remain, the studies in question will be resolved by discussion with a third reviewer from the project team.

In line with realist review methodology, iterative and purposive searches will be guided by the need to find more evidence to develop and test certain subcomponents of the programme theory. The project team will discuss and set the eligibility criteria for each additional search.

## Step 3: Extracting and organising data

The extraction and organising of data will be undertaken by one reviewer. A random subsample of data extraction will be cross-checked by another member of the research team for consistency. Any disagreements will be resolved by discussion. The main project team will interject to resolve disagreements when necessary.

Full-texts of eligible evidence will be uploaded into NVivo version 2020 [54]. NVivo is a qualitative data management tool that facilitates data organisation. The relevant sections of text will be coded relating to contexts, mechanisms and/or their association with outcomes. The approach will be both inductive (the creation of new conceptual labels based on the data) and deductive (coding that maps on to the conceptual labels the initial theory was based on). Iterative alternation between analysis of particular approaches and consultation with topic experts at key stages for sense-checking will be conducted during this step. The coding will follow a realist, explanatory logic starting from relevant outcomes. Attempts will then be made to interpret and explain how healthcare professionals respond to resources provided to them (the mechanisms) from different approaches aimed at reducing restrictive practices. The specific contexts or circumstances will then be identified when these mechanisms are likely to be triggered. If appropriate, each new aspect of data will be used to refine the programme theory. As refinement of the programme theory progresses, the included studies will be re-examined to search for relevant data that may have been initially missed. An overview of included studies will be provided by extracting key study characteristics (e.g. study design, key findings, type of approach used to prevent or reduce restrictive practices) separately onto an Excel spreadsheet that will be validated by consulting the main project team.

## Step 4: Synthesising the evidence and drawing conclusions

A realist logic of analysis will be applied that focuses on how the evidence supports, refutes, or provides alternative explanations for approaches in preventing or reducing the use of restrictive practices.

The process of evidence synthesis will be achieved using the following three-stages analytic processes [44]: i) juxtaposition of data sources; ii) reconciling contradictory data; and iii) consolidation of sources of evidence. The first stage will involve comparing and contrasting between data presented in different studies. For instance, a rich qualitative study that provides insights into how a certain outcome is achieved as described in a quantitative study. The second stage will involve examining results that differ in seemingly similar circumstances; seeking explanations for the different outcomes with a particular focus on contexts. The third step will involve making judgements whether similarities between findings presented in different sources are adequate to form patterns in the developing context-mechanism-outcome configurations (CMOCs) and programme theory. These processes will facilitate in making sense of the CMOCs and overarching programme theory, reducing the number of CMOCs by consolidation, and highlighting nuances that may be act as an avenue of further exploration, if necessary.

The analysis and synthesis stage of the review is an iterative process and the intent is to understand which mechanisms are triggered in different contexts as described within the studies in the review. Further iterative searching for data may be required at this stage to test particular subcomponents of the programme theory, where evidence may be lacking.

Finally, the refined theory will be used to develop recommendations for improving practices aimed at preventing and reducing restrictive practices in learning disabilities settings.

## Involving experts by experience in the review

The relevance and development of the review has been and will be sense checked with experts by experience (e.g. service users and carers) to ensure that it is consistent with the experiences and practices in UK healthcare settings. Three members of the research team, an advocate of learning disabilities, who is the co-founder of *Learning Disability England* and a carer who is a founder of the *Positive and Active Behaviour Support Scotland* network were consulted in the development of the protocol, as was a practitioner with extensive patient and public involvement experience. These co-investigators will be leading the consultation with three experts by experience groups during the review: two established for service users and one for carers. The experts by experience members' views will be sought during the review to: i) sense check emerging programme theory; ii) inform the search strategy; and iii) shape the terminology and language that is used throughout the review, to ensure that information is appropriate and accessible for a lay audience. They will play a key role in developing and delivering a grass root dissemination strategy.

## Ethical considerations and declarations

The study was approved by Manchester Metropolitan University, Health, Psychology and Social Care Research Ethics and Governance Committee on 30th October 2020 (approval number: 22510) prior to commencing any consultation and data collection.

## Review timing and data availability

The current review stage includes performing the CLUSTER approach on all eligible articles identified from the electronic database searches. It is expected that the team will finalise the review and produce a final report to be published by the NIHR by December 2022. The report will summarise the results from the review, presenting the refined programme theory and outlining recommendations for healthcare teams and organisations implementing approaches to prevent and reduce restrictive practices. Data will be made available upon study completion in keeping with the PLOS Data Policy.

## Discussion

### Novelty of the review

The use of restrictive practices for vulnerable people in mental health and learning disabilities settings is a continuing pressing issue both nationally and internationally. It is essential that healthcare professionals use appropriate approaches to prevent and reduce such practices. The literature has so far focused on the effectiveness and the impact of some of these approaches on reducing the use of restrictive practices, without considering underlying processes and contextual influences. The findings of this realist review have the potential to provide an evidence-base for how and why certain components of or certain approaches work and in what circumstances. Although there is a plethora of different kind of systematic reviews in this field, the mechanisms underlying their efficiency are unknown. This will be the first realist review to be undertaken on this topic and integrating the views, experiences and expertise of people with lived experience (patients and carers), professionals and practitioners in this field, academics and topic experts.

### Impact and dissemination

The results from the review will be used to inform future policy, research and practice in in this area. The research team, advisory panel and experts by experience groups will share findings through their networks and promote change beyond the end of the project.

In addition to producing a report which will be published by the National Institute of Health Research (NIHR), the findings of this realist review will also be made public through a peer-reviewed open access publication. In addition, to increase the visibility and impact of the findings, the dissemination strategy will build upon the participatory nature and involvement from our stakeholder group, including experts by experience. As such, findings will be disseminated and shared through knowledge exchange with stakeholders and policymakers at a national and international level via conferences and personal communication. Relevant regional, national and/or international conferences may include the International Association for the Scientific Study of Intellectual Disabilities, Restraint Reduction Network, and the British Institute of Learning Disabilities. Key stakeholders within the project and wider team will be consulted to disseminate findings through their local and national networks including Learning Disability Partnership Boards, Positive and Active Behaviour Support Scotland, and the Care Quality Commission. To increase the accessibility of the review findings, user-friendly summaries will be produced and tailored suitable for healthcare professionals, service users and their families. The use of social media platforms (e.g. Twitter, Blogs, and Podcasts) will be considered to increase engagement from the wider population.

### Limitations

It should be acknowledged that a realist review is entirely based on secondary data and that there may be gaps in the literature that a realist evaluation can hope to fill. There is always a risk regarding the plausibility of the emerging theory and that the evidence will not be sufficient to support this. In this case, we will highlight theories or mechanisms that need to be tested with further, more robust research, e.g. randomised trials.

Consulting with experts by experience, i.e. service users and carers, to support the development of the programme theory will bring some challenges. However, the following strategies have been developed to address any emerging issues: firstly, workshops to improve awareness and understanding regarding the realist review methodology will be organised; secondly, existing service users and carers groups will be used to support those who are less familiar with

research. The research team have already made contact with key carers, practitioners and experts by experience who are supportive of the research agenda and methodology.

Finally, it is acknowledged that the realist review will require time and commitment from stakeholders. The research team is well connected with key organisations in this area and, given the importance of this agenda, the team has been successful in attracting key and highly motivated co-applicants and members to the advisory panel and the experts by experience groups. Furthermore, given the size of the team and the various groups, any unexpected problems with availability can be managed and shared accordingly.

With regards to impact on practice, the current restructuring of services where people with learning disabilities access care, as well as the significant funding cuts to service providers in this area will be a challenge. A key element in stakeholder engagement events will therefore be to engage a wider and influential audience with the research findings and proactively facilitate a two-way conversation about barriers to implementation and how these can be overcome.

## Supporting information

**S1 Checklist. PRISMA-P 2015 checklist.**
(DOCX)

**S1 File. Protocol approved by ethics committee.**
(PDF)

## Author Contributions

**Conceptualization:** Alina Haines-Delmont, Kirstine Szifris, Elaine Craig, Joy Duxbury.

**Funding acquisition:** Alina Haines-Delmont, Elaine Craig, Joy Duxbury.

**Methodology:** Alina Haines-Delmont, Anthony Tsang, Kirstine Szifris, Elaine Craig.

**Project administration:** Alina Haines-Delmont, Joy Duxbury.

**Writing – original draft:** Alina Haines-Delmont, Anthony Tsang, Kirstine Szifris, Elaine Craig, Melanie Chapman, Joy Duxbury.

**Writing – review & editing:** Alina Haines-Delmont, Anthony Tsang, Kirstine Szifris, Elaine Craig, Melanie Chapman, John Baker, Peter Baker, James Ridley, Michaela Thomson, Gary Bourlet, Beth Morrison, Joy Duxbury.

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
