## [Decision Letter · Decision Letter 0]

13 Oct 2021

PONE-D-21-14394

Approaches used to prevent and reduce the use of restrictive practices on adults with learning disabilities: protocol for a realist review

PLOS ONE

Dear Dr. Haines-Delmont,

Thank you for submitting your manuscript to PLOS ONE. After careful consideration, we feel that it has merit but does not fully meet PLOS ONE’s publication criteria as it currently stands. Therefore, we invite you to submit a revised version of the manuscript that addresses the points raised during the review process.

The manuscript has been evaluated by one reviewer, and his comments are available below.<o:p></o:p>

The reviewer has raised a number of concerns that need attention. In particular, he requests to restructure the introduction to better clarify the general approach, and the inclusion and exclusion criteria. Moreover, he requests additional information on methodological aspects of the study (such as the inclusion of some parts of the “patient and public involvement" section in the Methods). 

background:#E6E6E6"><o:p></o:p>

Could you please revise the manuscript to carefully address the concerns raised?

background:white"><o:p> </o:p>Please submit your revised manuscript by Nov 27 2021 11:59PM. If you will need more time than this to complete your revisions, please reply to this message or contact the journal office at plosone@plos.org. Please include the following items when submitting your revised manuscript:

We look forward to receiving your revised manuscript.

Kind regards,

Lorena Verduci

Academic Editor

PLOS ONE

Journal Requirements:

Additional Editor Comments (if provided):

Reviewers' comments:

Reviewer's Responses to Questions

**Comments to the Author**

1. Does the manuscript provide a valid rationale for the proposed study, with clearly identified and justified research questions?

Reviewer #1: Partly

2. Is the protocol technically sound and planned in a manner that will lead to a meaningful outcome and allow testing the stated hypotheses?

Reviewer #1: Yes

3. Is the methodology feasible and described in sufficient detail to allow the work to be replicable?

Reviewer #1: Yes

4. Have the authors described where all data underlying the findings will be made available when the study is complete?

Reviewer #1: Yes

5. Is the manuscript presented in an intelligible fashion and written in standard English?

Reviewer #1: Yes

6. Review Comments to the Author

You may also provide optional suggestions and comments to authors that they might find helpful in planning their study.

Reviewer #1: As a protocol, it is therefore obvious that this type of paper has less impact on the scientific community. That being said, it is a solid and well-argued protocol.

I can easily identify two core strengths in the proposed protocol. The first one is the integration of “experts by experience”, healthcare professionals, and associations, that is coherent and well-justified. The second one is the need for a realist review in the field of seclusion and restraint prevention, regardless of the population. Although there is a plethora of different kind of systematic reviews in this field, we don’t know the mechanisms underlying their efficiency.

My main concern relates to the targeted population, adults with learning disabilities. In the introduction, there seems to be a blurring and constant back and forth between articles referring to interventions in a psychiatric context and articles referring to interventions in a context of learning disabilities. So I believe that the introduction should be restructured and deepened, which will allow a better understanding of the approach, and the inclusion and exclusion criteria.

The abstract is clear.

Title: “approaches” is preferred above programmes and interventions. One concept should be chosen and used throughout the paper.

Team: An experienced team is leading the project with many different backgrounds, which is a strength.

Introduction: The third paragraph (70-82) is weaker and needs to be better integrated. The authors refer to “restrictive practices in mental health or learning disability settings”, but differences and similarities amongst them should be explained.

83-85: “focused on their effectiveness on the reduction of behaviour that challenges, not necessarily the reduction of use of restrictive practices”: I disagree with this, especially as the authors base their arguments on three papers from the same research team.

86: The previous paragraph introduces different approaches, but not "multimodal programs" when indeed, a quick review of the literature can identify evidence to this effect. This is an example where I think the authors can restructure the introduction.

Some parts of the “patient and public involvement” section should be in the Methods. The need for patient and public involvement would benefit from being linked to the introduction, but the explanation of this kind of involvement in the realist review shoud be presented in the Methods section.

The section “Objectives” does not include objectives but the aim of the study, but this one is clear.

Design: The design is well-presented and described.

143-144: What is the link between reference 44 and the sentence? This reference should be in the introduction.

My main concern, as stated before, is on the targeted population: for step 1, is the focus of search goes “beyond specific interventions” as well as populations? It is not clear what will be specific to the targeted population.

For step 2, what is the process for the grey literature search? How will unpublished evidence be obtained?

188-195: Pawson’s method was first published in 2005, and the science of knowledge syntheses has evolved enormously since then. Although according to the realist review approach, the quality assessment with validated tools is not mandatory, several researchers are now proposing to carry out a quality assessment even if it was not previously required in the method. I suggest the authors refer to JBI’s Critical Appraisal Tools.

Some steps need more precision. For example, is the eligibility of evidence assessment based on the full paper, abstract, etc…

Step 3: can the authors be more precise on how they will code something like the “mechanisms”, since it’s “usually hidden”?

I’m not sure if the study’s timeline presented was adjusted.

7. PLOS authors have the option to publish the peer review history of their article (what does this mean?). If published, this will include your full peer review and any attached files.

Reviewer #1: **Yes: **Marie-Hélène Goulet

---

## [Author Response · Author response to Decision Letter 0]

12 Nov 2021

Dear reviewer and editor,

Many thanks for providing your feedback regarding our protocol. With this resubmission, we have included a letter responding to each point you have raised part of the peer review. We are really grateful for considering our protocol for publication and thank you for identifying both strengths and weaknesses which we have taken into account. Our response is detailed in the submitted 'Response to Reviewers' document and the 'Revised Manuscript' reflects the proposed changes.

In particular, (1) we have restructured the introduction to clarify the prevalence of use of restrictive practices particularly for people with learning disabilities and the type and scope of approaches used to reduce such practices for people with learning disabilities. This is supported by additional evidence (new citations and references). This strengthens the rationale and eligibility criteria for the review. W have also (2) included additional information on methodological aspects of the study, the role of patient and public involvement and the stages of the review.

Our response to the specific comments are as follows:

C1: My main concern relates to the targeted population, adults with learning disabilities. In the introduction, there seems to be a blurring and constant back and forth between articles referring to interventions in a psychiatric context and articles referring to interventions in a context of learning disabilities. So I believe that the introduction should be restructured and deepened, which will allow a better understanding of the approach, and the inclusion and exclusion criteria. Thank you for pointing out this limitation. 

R1: We have restructured the introduction to clarify the prevalence of use of restrictive practices particularly for people with learning disabilities and the type and scope of approaches used to reduce such practices for people with learning disabilities, supported by additional evidence (new citations and references). This strengthens the rationale for the review. See change in text and new citations lines: 60-62 (p. 3), 66, 72-79 (p. 4), 94-130 (pp. 5-6) in the revised manuscript with track changes. Some references were deleted (mental health specific) and new ones included (LD/ID specific).

C2: Title: “approaches” is preferred above programmes and interventions. One concept should be chosen and used throughout the paper. 

R2: This terminology (approach/approaches) is now used across the review protocol. 

C3: Introduction: The third paragraph (70-82) is weaker and needs to be better integrated. The authors refer to “restrictive practices in mental health or learning disability settings”, but differences and similarities amongst them should be explained.

R3: As specified above, we have now identified the key approaches used in these settings and summarised the evidence to support these. A parallel with evidence re mental health setting has been drawn.

C4: 83-85: “focused on their effectiveness on the reduction of behaviour that challenges, not necessarily the reduction of use of restrictive practices”: I disagree with this, especially as the authors base their arguments on three papers from the same research team.

R4: This paragraph has now been deleted and rephrased in line with the evidence.

C5: 86: The previous paragraph introduces different approaches, but not "multimodal programs" when indeed, a quick review of the literature can identify evidence to this effect. This is an example where I think the authors can restructure the introduction.

R5: This paragraph has now been rephrased in line with the evidence.

C6: Some parts of the “patient and public involvement” section should be in the Methods. The need for patient and public involvement would benefit from being linked to the introduction, but the explanation of this kind of involvement in the realist review shoud be presented in the Methods section.

R6: This has now been addressed. The section on inclusion of experts by experience/people with lived experience, i.e. patients and carers has now been moved at the end of the Methodology section (lines 309-322, p. 14 in the revised manuscript with track changes). A paragraph highlighting the benefits of including the views of experts by experience has been included at the end of the rationale section (lines 134-137, p. 6).

C7: The section “Objectives” does not include objectives but the aim of the study, but this one is clear.

R7: The title of the section has now been renamed ‘aim of the review’. If needed, we can include specific objectives, but we feel the main aim should suffice.

C8: Design: The design is well-presented and described.143-144: What is the link between reference 44 and the sentence? This reference should be in the introduction.

R8: The citation was erroneously included. This has now been removed.

C9: My main concern, as stated before, is on the targeted population: for step 1, is the focus of search goes “beyond specific interventions” as well as populations? It is not clear what will be specific to the targeted population.

R9: Lines 199-203 have now been deleted to prevent confusion.

C10: For step 2, what is the process for the grey literature search? How will unpublished evidence be obtained? 188-195: Pawson’s method was first published in 2005, and the science of knowledge syntheses has evolved enormously since then. Although according to the realist review approach, the quality assessment with validated tools is not mandatory, several researchers are now proposing to carry out a quality assessment even if it was not previously required in the method. I suggest the authors refer to JBI’s Critical Appraisal Tools. Some steps need more precision. For example, is the eligibility of evidence assessment based on the full paper, abstract, etc… Step 3: can the authors be more precise on how they will code something like the “mechanisms”, since it’s “usually hidden”?

R10: CLUSTER will be used to attempt to identify grey literature by contacting authors of eligible studies and searching on institutional repositories. Additionally, a free hand search on ProQuest and OpenGrey will be considered if CLUSTER yields insufficient grey literature. We have stated that quality appraisal of eligible articles will not be carried out as it is anticipated the search will yield opinion pieces, editorials and other forms of evidence that cannot be appraised using traditional quality assessment tools, which includes JBI’s tools. Additionally, it is important to acknowledge that we’re not including data based on the assessment of the methodology in which determined a particular effect size. In realist reviews, it is anticipated we’re only going to be using little bits of information from a variety of sources that cannot be formally assessed. For example, using authors’ interpretations. Evidence in realist reviews are typically judged on two domains: relevance and rigour. We have embellished this particular section in how specifically we’re categorising relevance and rigour (lines 258-268). Eligibility of articles will be assessed at two stages: title/abstract and then at full-text level (line 275). 

To address the comment relating to coding the mechanisms, it is important to acknowledge that this process is interpretative. We have therefore added the following sentences to reiterate this: “The coding will follow a realist, explanatory logic starting from relevant outcomes. Attempts will then be made to interpret and explain how healthcare professionals respond to resources provided to them (the mechanisms) from different approaches aimed at reducing restrictive practices. The specific contexts or circumstances will then be identified when these mechanisms are likely to be triggered. ” (lines 292-299)

C11: I’m not sure if the study’s timeline presented was adjusted. 

R11: The timeline of this review has not changed and is still expected to run for a 22-month period from September 2020. The searches, however, were conducted in July instead of May. This change has now been reflected in the main text (line 221).

Hope our repose satisfies the reviewer's request for clarification and additional information.

Kind regards,

Alina Haines-Delmont

---

## [Decision Letter · Decision Letter 1]

3 Jun 2022

Approaches used to prevent and reduce the use of restrictive practices on adults with learning disabilities: protocol for a realist review

PONE-D-21-14394R1

Dear Dr. Haines-Delmont,

We’re pleased to inform you that your manuscript has been judged scientifically suitable for publication and will be formally accepted for publication once it meets all outstanding technical requirements.

Kind regards,

Sara Rubinelli

Academic Editor

PLOS ONE

Additional Editor Comments (optional):

Reviewers' comments:

Reviewer's Responses to Questions

**Comments to the Author**

1. Does the manuscript provide a valid rationale for the proposed study, with clearly identified and justified research questions?

Reviewer #1: Yes

2. Is the protocol technically sound and planned in a manner that will lead to a meaningful outcome and allow testing the stated hypotheses?

Reviewer #1: Yes

3. Is the methodology feasible and described in sufficient detail to allow the work to be replicable?

Reviewer #1: Yes

4. Have the authors described where all data underlying the findings will be made available when the study is complete?

Reviewer #1: Yes

5. Is the manuscript presented in an intelligible fashion and written in standard English?

Reviewer #1: Yes

6. Review Comments to the Author

You may also provide optional suggestions and comments to authors that they might find helpful in planning their study.

Reviewer #1: The authors have precisely responded to the questions I had previously raised. The protocol proposes a method less used in this research field and will allow answering the questions raised by the current state of knowledge. I will read the results of this study with pleasure, as I believe they will potentially influence future research and practices.

7. PLOS authors have the option to publish the peer review history of their article (what does this mean?). If published, this will include your full peer review and any attached files.

Reviewer #1: **Yes: **Marie-Hélène Goulet

---

## [Editor Report · Acceptance letter]

7 Jun 2022

PONE-D-21-14394R1 

Approaches used to prevent and reduce the use of restrictive practices on adults with learning disabilities: protocol for a realist review 

Dear Dr. Haines-Delmont:

I'm pleased to inform you that your manuscript has been deemed suitable for publication in PLOS ONE. Congratulations! Your manuscript is now with our production department. 

Kind regards, 

on behalf of

Dr. Sara Rubinelli 

Academic Editor

PLOS ONE